# Re_Trans: Combined Retrieval and Transformer Model for Source Code Summarization

**DOI:** 10.3390/e24101372

**Published:** 2022-09-27

**Authors:** Chunyan Zhang, Qinglei Zhou, Meng Qiao, Ke Tang, Lianqiu Xu, Fudong Liu

**Affiliations:** 1State Key Laboratory of Mathematical Engineering and Advanced Computing, Zhengzhou 450001, China; 2School of Information Engineering, Zhengzhou University, Zhengzhou 450001, China

**Keywords:** source code summarization, program analysis, information retrieval, deep learning

## Abstract

Source code summarization (SCS) is a natural language description of source code functionality. It can help developers understand programs and maintain software efficiently. Retrieval-based methods generate SCS by reorganizing terms selected from source code or use SCS of similar code snippets. Generative methods generate SCS via attentional encoder–decoder architecture. However, a generative method can generate SCS for any code, but sometimes the accuracy is still far from expectation (due to the lack of numerous high-quality training sets). A retrieval-based method is considered to have a higher accurac, but usually fails to generate SCS for a source code in the absence of a similar candidate in the database. In order to effectively combine the advantages of retrieval-based methods and generative methods, we propose a new method: Re_Trans. For a given code, we first utilize the retrieval-based method to obtain its most similar code with regard to sematic and corresponding SCS (S_RM). Then, we input the given code and similar code into the trained discriminator. If the discriminator outputs onr, we take S_RM as the result; otherwise, we utilize the generate model, transformer, to generate the given code’ SCS. Particularly, we use AST-augmented (AbstractSyntax Tree) and code sequence-augmented information to make the source code semantic extraction more complete. Furthermore, we build a new SCS retrieval library through the public dataset. We evaluate our method on a dataset of 2.1 million Java code-comment pairs, and experimental results show improvement over the state-of-the-art (SOTA) benchmarks, which demonstrates the effectiveness and efficiency of our method.

## 1. Introduction

Source code summarization (SCS), also named code comment, is a term coined by Haiduc et al. [1]. It is a natural language description of programming fragments. Program maintenance is the most expensive and time-consuming stage in the software life cycle [2]. High-quality SCS is essential to program comprehension and maintenance, which can help developers save time spent on navigating source code and understand programs quickly. Unfortunately, with the rapid update of software, most SCS is mismatched, outdated, and missing. Hence, SCS generation has been researched extensively and has made lots of remarkable achievements [3,4,5,6,7,8,9,10,11,12,13].

SCS generation is a hot field that emerged more than a decade ago. Its methods can be divided into three categories: manually-crafted template, information retrieval-based (IR-based), and deep-learning-based (DL-based). The manual template methods usually extract keywords from source code to generate SCS [4,14,15]. However, they miss a lot of potential information of the source code. The IR-based methods are widely used in SCS generation. They usually generate SCS by searching keywords from the given code or code comments of the code that are most similar to the given code. For example, Haiduc et al. [1,3] analyzed source code using the vector space model (VSM) and latent semantic indexing (LSI) methods, producing natural language description of the classes or methods. Li et al. [16] used the latent dirichlet allocation (LDA) technology to conduct topic mining on resources, such as code, documentation, question and answer information, and automatically generated code topic summarization. Wong et al. [17] utilized code clone detection technology to find the code snippet with similar syntax from the existing code bases and applied summarization to other codes with similar syntax. However, IR-based methods over-rely on identifier naming and the similar amount of source code in the dataset.

Currently, almost all works use the DL-based method in SCS generation task. The common DL techniques include recurrent neural network (RNN) [18] and its variant models, convolution neural network (CNN) [19] and its variant models, transformer model [20], and large-scale language training model (e.g., BERT [21] and GPT [22]). The attention mechanism is usually used as a key auxiliary to the above methods. Deep-Com [8] utilized a seq2seq model to generate SCS of a Java method based on the attention mechanism, and its SBT method (a tree traversal way) made a major breakthrough in structure information extraction. Notably, the SBT method has been adopted by many works. For example, in 2020, the Hybrid-DeepCom [23] extended the work of paper [8]. It combined the source code sequence and the SBT sequences to generate SCS. Especially, the camel case naming was used to solve the out-of-vocabulary identifiers problem. LeClair et al. [24] improved the accuracy of SCS by processing source code AST information on the basis of ast-attendgru [10]. Wang et al. [12] and Uddin Ahmad et al. [13] utilized transformer to generate the SCS, and improved the effectiveness and accuracy compared with existing methods. The quality of code summarization generated by the methods in papers [13,24] is better than the other methods mentioned above. The main reason is that the paper [24] takes the whole AST as a graph to represent structure information instead of AST-sequences or AST-paths, which preserves the structure information more completely. Hence, we use this AST embedding way in our method. Moreover, the paper [13] proves that the transformer model performs well in the SCS generation task. However, these methods use either IR-based or neural machine translation (NMT)-based methods to generate SCS. NMT-based methods are generative methods. A generative methord can generate SCS for any code, but the result is still far from expectation due to the absence of a high-quality training set. A retrieval-based method has high accuracy, but it requires a similar candidate in the database to the given code.

In this paper, for the purpose of combining the advantages of retrieval-based methods and generative methods, we propose a neural approach to generate SCS, Re_Trans. It contains one retrieval-based model and one generative model and uses a discriminator to decide which model’s result is the final SCS for the give code. For a given code, we first utilize a retrieval-based method to obtain the most similar code with regard to semantics and its SCS (S_RM). Then, we input the given code and similar code into the trained discriminator. If the output is one, we take S_RM as the result; otherwise, we utilize the transformer model to generate the final SCS. Re_Trans adopts a suitable SCS generation model to the given code.

In particular, we propose a new method that combines the enhanced code sequences and enhanced code structures to represent the source code semantic, which are implemented as follows: (1) We use AST to represent the structure information and enhance it by adding data flow and control flow edges to AST. Moreover, we utilize a graph convolutional network (GCN) [25] to encode the whole AST for preserving the structure information more completely. (2) We use code sequence to represent the syntax information and enhance it by adding position information to code. In retrieval model, we adopt a bidirection gate recurrent unit (BiGRU) [26] to encode code sequences and choose a self-attention mechanism to encode them in a transformer model.Moreover, we use a beam search algorithm [27] in Re_Trans to ensure that the generated SCS is non-random and closest to the real result. We conduct experiments on a popular real-word dataset, and the results demonstrate that our method outperforms the SOTA work (in Section 3.3) with widely-used metrics (BLEU, METEOR, and ROUGE) in code summarization tasks. Furthermore, we also perform time-consuming experiments to confirm the efficiency of our method.

The main contributions of this paper are as follows:We propose a Re_Trans system by combining retrieval and generative methods and adopt the suitable SCS generation model for a given source code.We use non-leaf nodes of the AST to build a directed graph and enhances the edge information thought data flow and control flow. To the best of our knowledge, this is the first time that such an efficient structure representation mode has been used in an SCS task.We perform extensive experiments on a public real-world dataset. All results confirm that the Re_Trans is effective and outperforms the SOTA methods.

## 2. Our Approach

### 2.1. Overview

The workflow of our proposed method (Re_Trans) is shown in Figure 1.

Re_Trans mainly contains three steps: (1) Data representation (see Section 2.2): Re_Trans parses the source code into AST and source code sequence and processes code summarization by a plain text that is composed of tokens (i.e., variables). (2) Model training design (see Section 2.3); Re_Trans includes a retrieval-based model, a generative model, and a discriminator (see Section 2.5). (3) Model test design (see Section 2.4).

### 2.2. Data Processing

In this paper, we use large public dataset <Java code, comment> pairs. Our data processing method is available in various programing languages. We represent the Java code as parsed AST and code sequence and process comments into plain text.

For one sample, we show the source code structure information in Figure 2. Initially, we use the javalang (https://github.com/kangyujian/JavaMethodExactor, accessed on 20 August 2020) toolkit to parse source code into an AST and remove the leaf nodes of AST. There are two reasons for removing the leaf nodes: (1) To avoid repeated processing because the leaf nodes correspond to the source code text, which has been processed in code sequence information. (2) Non-leaf nodes represent the source code structure information to a certain extent, and this structure saves much traversal time.

Furthermore, we enhance source code semantic information by adding data flow and control flow to AST referring to the work of Wang et al. [28]. Considering our AST without leaf nodes, in data flow information, we connect a node to its next brother node (from left to right). It solves the problem that graph neural networks do not consider the order of nodes. For example, the green arrows in the red dotted box (aa) in Figure 2 connect three sibling nodes of “Modifier”, “FormaParameter”, and “ForStatement”. The added edges are (Modifier, FormaParameter), (FormaParameter, and ForStatement). In control flow information, we select “IfStatement”, “WhileStatement”, “ForStatement”, and “BlockStatement” nodes. “BlockStatement” is the root node of the code block when executing source code sequentially. According to the characteristics of each node, we connect their child nodes to form new edges.

Finally, we utilize depth-first traversal to obtain the edge set of the directed AST. The edge set is $e=(e_{1},e_{2}\ldots e_{n})$, where *n* is the number of edges. Compared with undirected AST, directed AST can represent the sequence structure information more accurately. We take the edge information as an initialization vector and input it into GCN for semantic extraction of the source code.

The source code sequence information is shown in Figure 3. Firstly, we treat each source code as a plain text (as shown in the blue box); each word corresponds to a unique identifier (token id) by dictionary mapping. Then, we add row and column position information to the source code. For the row position information, we assign integer values starting from zero to each row of Java function sequentially (as shown in the red box). For the column position information, we assign integer values starting from zero in the word order of each code (as shown in the green box). Hence, each word in the source code sample has three feathers: token id, column position, and row position. We concatenate the initialized vectors of these features as the initial vector for each word. In this way, we complete the vector initialization of the source code and utilize the self-attention mechanism to obtain the source code sequence sematic vector. We take “static” as an example. From Figure 3, we can see that its token id is 35, column position is 1, and row position is 0. Firstly, we perform vector initialization on 35, 0, and 1, respectively, and obtain the corresponding embeddings $Emb_{1}$, $Emb_{2}$, and $Emb_{3}$. Secondly, we concatenate the three embeddings to obtain the initial vector of “static”, $Emb=cat(Emb_{1},Emb_{2},Emb_{3})$. At last, we input Emb into the self-attention mechanism to obtain the final semantic vector of “static”. Let the word dimension be *d*, then $Emb_{1},Emb_{2},Emb_{3}\in R^{1\times d}$, $Emb\in R^{1\times 3d}$.

Code comment is similar to text in NLP, without complex structure information. We obtain their semantic vectors through the self-attention mechanism and use it as part of the decoder input of the transformer model.

### 2.3. Model Training Design

Re_Trans contains two SCS generation models: retrieval model and transformer. We show them in Figure 4 and Figure 5.

In the retrieval model, when comparing the codes’ plain texts, it is difficult to judge whether they are similar because different programming texts may implement the same function. For example, both “forstatement” and “whilestatement” can implement the loop function. The semantic similarity measures the difference between tokens based on the similarity of their meaning or semantic content rather than the similarity of dictionaries. It often uses statistical methods, such as vector space models, to associate words and textual contexts from corpora. Therefore, we compare the code’ semantic similarity in the retrieval-based model. For a sample, we use GCN to process the enhanced AST and BiGRU to deal with the enhanced code sequence. The semantic vector of a sample is the results concatenation of GCN and BiGRU, namely Emb_S. We use the n-dimensional Euclidean distance formula that is shown as (1) to find the code in the retrieval library that is most similar to the sample, namely Emb_Simi, and its SCS. The construction process of the retrieval library will be detailed in Section 3.1.
(1)$$\begin{array}{c} d_{\left(Emb\_S,Emb\_Simi\right)}=\sqrt{\sum_{i=1}^{n}{\left(x_{i}-y_{i}\right)}^{2}} \end{array}$$
where $Emb\_S\;=\;(x_{1},x_{2},\ldots ,x_{n})$, $Emb\_Simi\;=\;(y_{1},y_{2},\ldots ,y_{n})$, and i∈[1,n]

The goal of the transformer model is to generate a new SCS for each input function. For a sample, we also use GCN to extract its structure information. As for the syntactic information, we directly utilize the transformer’s encoder. The sample semantic vector is the results concatenation of GCN and the self-attention mechanism. We use the transformer’s decoder to convert the sample semantic vector to its SCS.

### 2.4. Model Test Design

In this section, we show the test flow in Figure 6. To make it easier to understand, we introduce the test flow through a running example that is shown in Figure 7. We first input the Java code (a) into the retrieval-based model introduced in Section 2.3 and obtain the semantic vector of Java code (b) and target summarization (b). Even more, (b) is the most similar Java code to (a). Then, we input the semantic vectors of (a) and (b) into the trained discriminator, and the sim_label is one, so we obtain the target summarization (b) as the final summarization result, which means it “generates the most likely state predictions for the sequence”. Specifically, the trained discriminator’s training process is detailed in Section 2.5.

### 2.5. Discriminator

In particular, it is stated that the role of discriminator in this paper is different from that of the discriminator in the adversarial generative network. The purpose of our proposed discriminator is to judge whether the Java code summarization produced by the retrieval model is optimal, which is used in both training and test phases. Furthermore, the discriminator’s working approach is detailed in its training and test process.

In order to train the discriminator, we randomly sample 200,000 samples from the dataset in Section 3.1 and divide them into training and testing sets in a ratio of 8:2. The training process is shown in Figure 8.

First, we assign the label to all samples. For a sample: (1) We use the retrieval model in Section 2.3 to obtain its S_code and S_RM and use the transformer model to generate its SCS (S_GM). 2) When analyzing the similarity between two feature vectors, the cosine similarity can avoid the large distance caused by different sequence lengths and only consider the angle between two vectors. Therefore, we utilize the cosine distance to calculate the similarity between S_RM, S_GM, and the target SCS (T_SCS), respectively, which are shown in Formulas (2) and (3), where $S\_RM=(s\_rm_{1},s\_rm_{2},\ldots ,s\_rm_{m})$, $T\_SCS=(c_{1},c_{2},\ldots ,c_{m})$, $S\_GM=(s\_gm_{1},s\_gm_{2},\ldots ,s\_gm_{m})$, and *m* represents the dimension of code summarization. If S_RM is better, we set the sample as <code, S_code, 1>. Otherwise, the sample is <code, S_code, 0>. We repeat these two steps, and we set all discriminator data in the form of <code, S_code, label>, where the label represents zero or one.
(2)$$\begin{array}{c} cos\_{\mathrm{simi}}_{r}=\frac{\sum_{i=1}^{m}s\_rm_{i}\cdot c_{i}}{\sqrt{\sum_{i=1}^{m}s\_rm_{i}^{2}\cdot \sqrt{\sum_{i=1}^{m}c_{i}^{2}}}},i\in \left[1,m\right] \end{array}$$
(3)$$\begin{array}{c} cos\_{\mathrm{simi}}_{g}=\frac{\sum_{i=1}^{m}s\_gm_{i}\cdot c_{i}}{\sqrt{\sum_{i=1}^{m}s\_gm_{i}^{2}\cdot \sqrt{\sum_{i=1}^{m}c_{i}^{2}}}},i\in \left[1,m\right] \end{array}$$

Second, we train the discriminator with 160,000 samples, and the parameters are shown in Section 3.2. Especially, we utilize Multilayer Perceptron (MLP) to calculate the semantic similarity between SCSs. Finally, we use the remaining 40,000 samples to test the trained discriminator.

## 3. Experiments Setup

### 3.1. Dataset Analysis

The dataset contains around 2.1 million <Java code, comment> pairs [29], which are widely used in lots of SCS generation tasks [10,20,30]. We analyzed the dataset from two aspects: (1) statistical length distribution of source codes and their comments (see Figure 9); and (2) a count of the scale of Java code numbers with the same comment (see Figure 10).

From Table 1 and Figure 9, we can see that the comment length distribution is relatively uniform, ranging from 3 to 13. A short comment helps people understand the code function quickly. The code lengths are distributed between 1 and 100, which is approximately normal distribution. When code length is larger than 70, the number is almost unchanged, so we set 70 as the optimal input length parameter.

In order to train a retrieval library, we remove invalid data whose comment corresponds to only one function. In Figure 10a, the function of scale two is close to 160,000, far exceeding the number of other scales. We denoise these functions and use them as a retrieval library. From Figure 10b, we find that the number of scales over 80 is almost 1.

### 3.2. Parameter Settings

In this section, we introduce the main parameter settings in all experiments as shown in Table A1. In the transformer model, we use the Adam optimizer and set epsilon to $1\times 10^{-9}$ and $\left(\beta_{1},\beta_{2}\right)$ to (0.9, 0.98). The parameters of thetransformer model are N = 4, h = 4, and dim = 256), where N is the number of encoder layers, h is the number of multi-head-attention, and dim is the embedding dimension. Particularly, we use the NoamOpt to obtain the learning rate dynamically, where the warmup is 200 and the factor is 1. The batch_size is set to 256, and the epoch is 40. In the retrieval model, we also use the Adam optimizer, and set the GCN and BiGRU layer to two. Moreover, we set the leaning rate to $1\times 10^{-4}$, and the epoch to 30.

We conduct all the experiments on a workstation with two Intel(R) Xeon(R) Gold 6154 CPU@3.00 GHz, 128 gb RAM, and two Titan XP GPUs. It is necessary to train on GPUs with 64gb VRAM due to the large size of our model and dataset.

### 3.3. Baselines

To demonstrate the effectiveness of our method, we compare it with the SOTA methods from recent years. The baselines are described as follows:

LeClair et al. [10] (2019) proposed a method called ast-attendgru that was an attentional encoder–decoder architecture to generate SCS. Ast-attendgru enhanced the SBT and AST flattening procedure proposed by Hu et al. [7,23] and showed a higher performance. Hence, we only compare against this approach.

Xu et al. [31] (2018) proposed an approach called graph2seq which was a general neural encoder–decoder architecture that solved the graph-to-sequence problem. It achieved SOTA results on an SQL-natural language task using BLEU-4 metric. The open-source code of graph2seq is convenient to conduct comparative experiments.

LeClair et al. [24] (2020) adopted code+gnn+BiLSTM to generate SCS. Different from the flattened AST, they took the AST as a graph, and it performed well, which was closer to our method in terms of code structure information extraction.

Ahmad et al. [13] (2020) proposed the transformer-based method that was the first to apply the transformer model to source code comments. They incorporated relative positional encoding and copied an attention mechanism into the transformer model to improve the SCS quality. We also use the transformer model as the Re_Trans’ generative model.

Zhang et al. [11] (2020) proposed a novel retrieval-based neural approach called Rencos. Rencos retrieved the two most similar code snippets in a given code from aspects of semantic and syntax, respectively. Rencos enhanced the accuracy of the SCS by fusing the retrieved results into the generative model.

Wei et al. [32] (2020) proposed an approach to enhance the SCS’ accuracy, namely Re2Com. Similar to Rencos, Re2Com also used a retrieval-based method to enhance the SCS’ accuracy. For a given code, Re2Com retrieved its most similar code and the SCS pair. Then it took the given code (its code text and AST sequence), the most similar code, and SCS as input to the encoder. Experiments demonstrated the effectiveness of this method.

## 4. Results and Analysis

Our research objective was to determine that the Re_Trans outperforms current baselines. We also wanted to demonstrate the efficiency of RGSGS and the effectiveness of the retrieval library built by us. Notably, all methods were trained on the dataset described in Section 3.1 and evaluated by the widely-used metrics BLEU [33], METEOR [34], and ROUGE [35], detailed in the Section A.2. The metrics’ scores were in the range [0, 1] and reported in percentages in this paper. We answer the following research questions (RQs) to explore these situations:

RQ1: What is the performance of Re_Trans compared to the baselines? RQ2: Why does the Re_Trans approach perform well? RQ3: Where is the high efficiency of Re_Trans reflected? RQ4: What is the quality of the SCS retrieval library we built?

### 4.1. Re_Trans vs. Baselines

Among baselines, ast-attendgru, graph2seq, code+gnn+BiLSTM, and transformer-based belong to GM. Rencos and Re2Com are methods that combine retrieval and generative techniques. According to the parameter settings in Table A1, we show the experimental results of Re_Trans and baselines in Table 2.

From Table 2, we find that the effect of the Rencos method is relatively poor. The reason may be that it does not matter whether the retrieved similar code is actually similar to the input one or not. When taking them as input of the encoder, Rencos may produce biased SCS. Although ast-attendgru, Rencos, and Re2Com use flatted AST to represent code structural information, the AST sequence is a linear problem in nature. The effect of code+gnn+BiLSTM, transformer-based and Re2Com are close to Re_Trans. The Re_Trans and code+gnn+BiLSTM use a similar source code semantic extraction method, AST graph, and source code sequence, but the Re_Trans performs better. One reason is that our method enhances the AST and code sequence, which can extract the source code semantic information more fully (explained in Section 4.2). LeClair et al. [30] mentioned that the decoder with an attention mechanism is less effective than a transformer. We also find that the BLEU-N of the transformer-based method is on average about 7% higher than code+gnn+BiLSTM. The poor performance of graph2seq is because it only considers the structural information of source code but ignores the syntactic information in the SCS task.

In Table 2, the Re_Trans performs best; the main reasons are the following: (1) We combine the code sequence-augmented and AST-augmented to characterize source code. (2) The generative model is the transformer model. Transformer is generally better than the seq2seq architecture in all tasks. In addition, the BLEU-N gradually decreases as the N increases. It shows that there is still a lot of room for improvement in the long sequence matching between the generated SCS and the target. Furthermore, it also indicates that the current SCS generation model still needs to be further studied and improved.

### 4.2. Ablation Study

An ablation study is often used to reduce some improved features on the model proposed in the paper in order to verify the necessity of corresponding improved features. Ablation is a very labor-saving way to study cause and effect. In this section, we will illustrate why the Re_Trans method works well through source code representation ablation experiments. The source code semantic representation methods mainly include source code sequence information (Seq), AST-augmented (ast_aug) combined with Seq, and AST-augmented combined with code sequence-augmented information (Seq_aug). However, we also test the SCS effect of preserving leaf nodes of AST-augmented (ast_leaf_aug) combined with Seq_aug. For these different semantic extraction methods of source code, we use Re_Trans to generate SCS and show the ablation experiments results in Table 3.

In Table 3, we find that the SCS quality has been improved after the Seq combining with the ast_aug information. It is because pure sequence information ignores the potential and complex structural information of source code, and ast_aug preserves the structural information of source code. As we all know, the self-attention mechanism ignores position information when encoding sequence information. Therefore, we add position information to source code sequences to solve this problem. As shown in Table 3, the effect is significantly improved when we use “ast_aug+Seq_aug” to represent source code. The BLEU-1 is improved by 10.2%, and BLEU-4 is improved by 9.9%. Moreover, we also find that the effect of “ast_leaf_aug+Seq_aug” is better than “ast_aug+Seq_aug”, but the gap is slight. In Section 4.3, we demonstrate that the time efficiency of the latter is much higher than that of the former. Therefore, we choose the AST without leaf nodes to characterize the source code structure information.

### 4.3. High Efficiency

High efficiency is an important advantage of Re_Trans compared to other SCS methods. It is mainly reflected in two aspects: the efficiency of source code semantic extraction and the efficiency of Re_Trans’ generation model.

(1)Efficiency of source code semantic extraction:

The AST contains a large number of leaf nodes with irregular user-defined identifiers, which makes the data processing time-consuming. From Table 3, we know that the SCS’ result of AST with leaf nodes (ast_with_leaf) is close to that without leaf nodes (ast_no_leaf) in our method. In order to demonstrate the efficiency of source code semantic extraction, we randomly select 100,000 samples from the dataset and construct AST structure information with and without leaf nodes, respectively. We calculate the accumulated time (the time unit is seconds) for each 10,000 samples, and show the results in Figure 11a. Furthermore, we randomly select 1000 samples from these 100,000 samples, and test their SCS time through Re_Trans for these two source code structure representations, respectively. We calculate the accumulated time (the time unit is seconds) for each 100 samples processed and show the results in Figure 11b.

In Figure 11a, the ast_no_leaf extraction method we proposed (the green line) takes significantly less time, and the growth rate is slow, which reflects the efficiency of AST without leaf nodes in data processing. In Figure 11b, our proposed AST structure (the purple line) is more efficient than the ast_with_leaf extraction method in the SCS task. Therefore, we use the AST that removes leaf nodes, which not only saves the time of graph traversal, but also avoids the repeated operation of source code sequence information processing.

(2)Efficiency of the Re_Trans generation model:

We randomly selected 1000 samples from the dataset. We test the SCS generation time of Re_Tran’ generation model (Re_Trans_generative) on these data, Re_Trans, Re_Trans generative model with leaf nodes (Re_Trans_leaf_generative), Rencos and Rencos generative model (Rencos_only_NMT). The time statistic results are shown in Figure 12.

From Figure 12, we can see that Re_Trans_generative (the purple line) takes the shortest time and is more efficient than Rencos_only_NMT (the cyan line). Moreover, the Rencos method (the blue line) takes the longest time, and the efficiency of our method (the green line) is higher than Rencos. The main reason is that Rencos requires both IR-based and NMT-based methods for each test. However, Re_Trans only uses the IR-based method, or the IR-based and generative methods for each test. The abovementioned generation model of Re_Trans is more efficient than Rencos. Furthermore, in our test, the Rencos retrieval database has 6648 samples and Re_Trans has 10,000 samples, but their average retrieval time of one test data is about 0.104 s. Obviously, the Re_Trans’ retrieval efficiency is higher. In practice, with the continuous expansion of the retrieval database, the average retrieval time will increase accordingly. Because the test data needs to match each item in the retrieval database to find the sample with the highest similarity score.

### 4.4. SCS Library

From the dataset analysis in Section 3.1, we find that the public dataset has a large number of similar functions, which are suitable for building an SCS retrieval library. In order to demonstrate the effectiveness of the retrieval library, we conduct the following experiments:

(1) In order to test the effectiveness of our retrieval library, we randomly select samples from the code retrieval library after data processing and use the t-SNE data dimensionality reduction and visualization technology. The final result is shown in Figure 13.

In view of the fact that too many samples lead to a large number of repeated points in the picture, which affects the visualization effect, we select two groups of 100 and 200 random samples corresponding to (a) and (b) in Figure 13. From Figure 12, we can see that almost all the two points with the same color overlap or are very close. It indicates that functions with the same summarization still retain the same semantic information after data processing. Furthermore, it also shows the effectiveness of a retrieval library built by us.

(2) In order to test the effectiveness of the Re_Trans retrieval model, we randomly select 1000 samples and calculate the probabilities of Topk (p@k) between test function and each sample by Euclidean distance, where *k* is 1, 3, 5, and 10. The Topk is to find the top *k* numbers from the retrieval library. The higher p@k, the better matching effect. The test is carried out in 10 groups, and we use the boxplot for visual display. We show the result in Figure 14.

From Figure 14, we can see that the p@k increases gradually with the increase of *k*. In 10 rounds of testing experiments, when k=3, the minimum value is 80%, the maximum value is 96%, and the average value is 92%. For any test code snippet, it indicates that the same semantics’ code snippets searched from the retrieval library are the similar code snippets with higher accuracy. It also indicates the feasibility of using the SCS of similar code as the SCS of the test code.

## 5. Conclusions and Discussion

In this paper, we combine AST-augmented and code sequence-augmented to represent source code semantic information. We propose an efficient and accurate SCS generation system, Re_Trans. It first utilizes a retrieval-based model to obtain the most similar code with regard to semantics and its SCS (S_RM). Then, it feeds the given code and its similar code to the trained discriminator. Finally, it decides to use the S_RM as a result or utilize the transformer model to obtain the new result according to the discriminator’ output. Moreover, we conducted a series of contrast and ablation experiments to demonstrate that the Re_Trans outperforms existing SOTA methods. Combined with the recent work and the research of this paper, we suggest some valuable research points for the future:

In the future, we plan to expand the SCS retrieval library and pay special attention to the quality of the expansion data. Furthermore, we also plan to further investigate the usefulness of our approach, using it to generate SCS for other program languages without code comments. Moreover, a large-scale language training model will be an inevitable requirement with the increasing daily data. Therefore, it is a meaningful research direction that includes extracting effective semantic information without occupying too many computing resources.

## Figures and Tables

**Figure 1 entropy-24-01372-f001:**
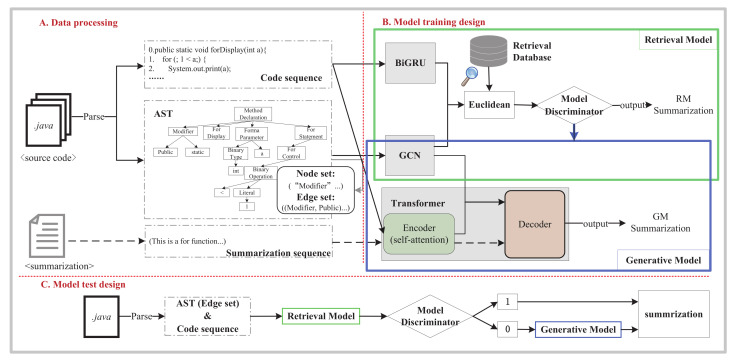
Overall framework of Re_Trans (color print).

**Figure 2 entropy-24-01372-f002:**
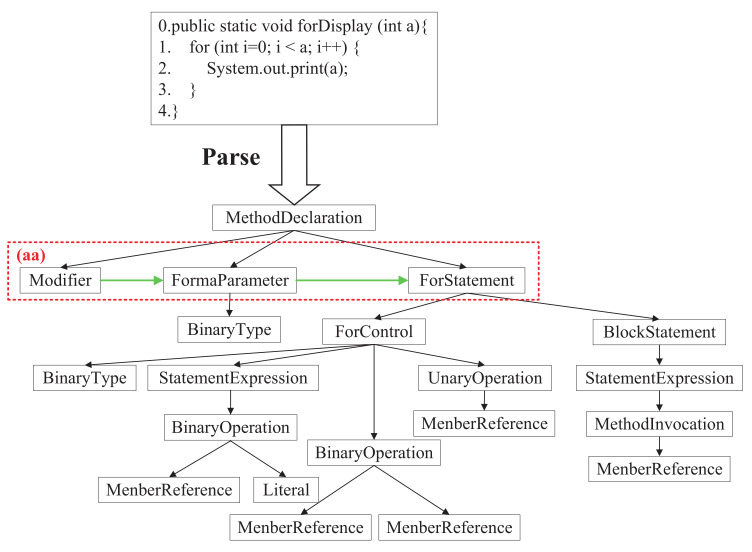
The parsed AST from source code (color print).

**Figure 3 entropy-24-01372-f003:**
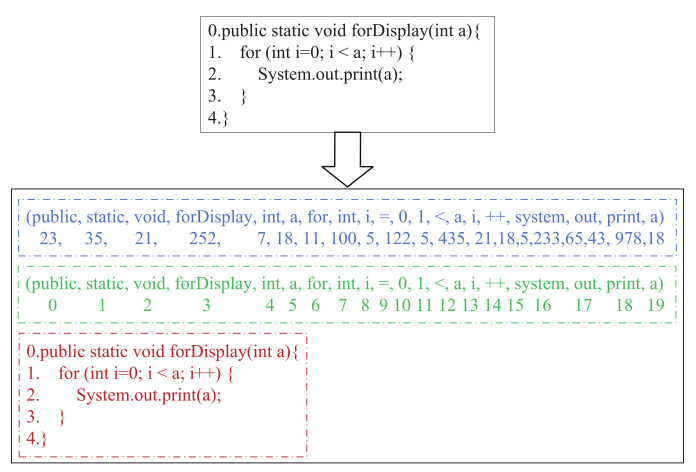
The code sequence information of source code (color print).

**Figure 4 entropy-24-01372-f004:**
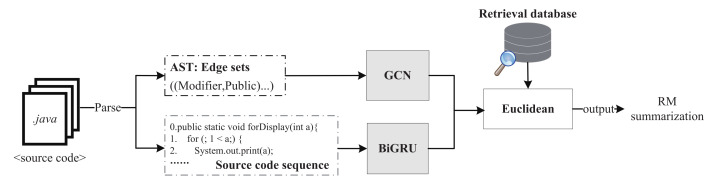
The retrieval-based model (color print).

**Figure 5 entropy-24-01372-f005:**
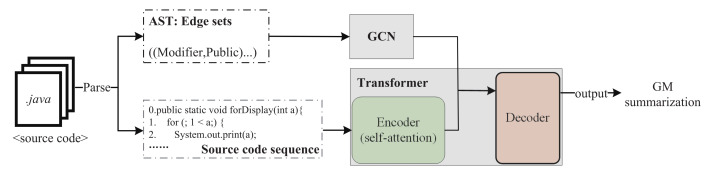
The generative model (color print).

**Figure 6 entropy-24-01372-f006:**
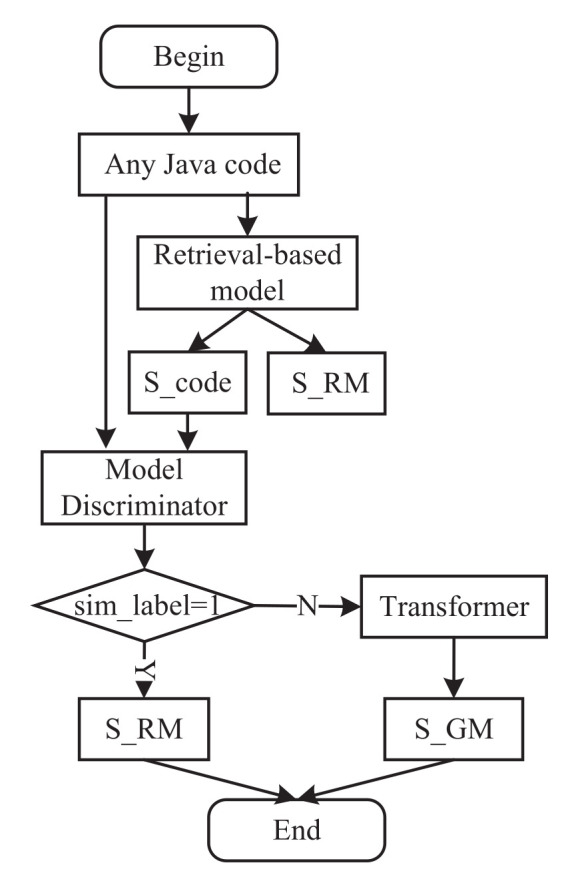
The test workflow of Re_Trans.

**Figure 7 entropy-24-01372-f007:**
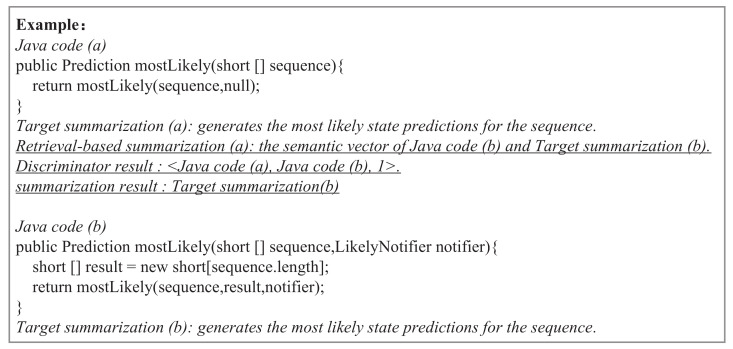
The running example.

**Figure 8 entropy-24-01372-f008:**
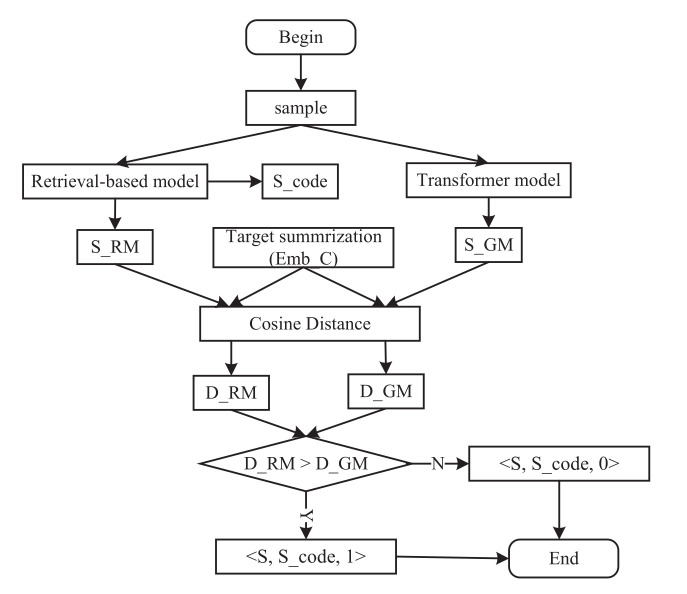
The training workflow of Re_Trans.

**Figure 9 entropy-24-01372-f009:**
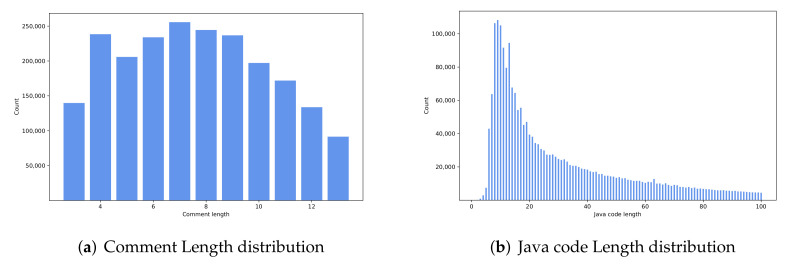
Length distribution of dataset (color print).

**Figure 10 entropy-24-01372-f010:**
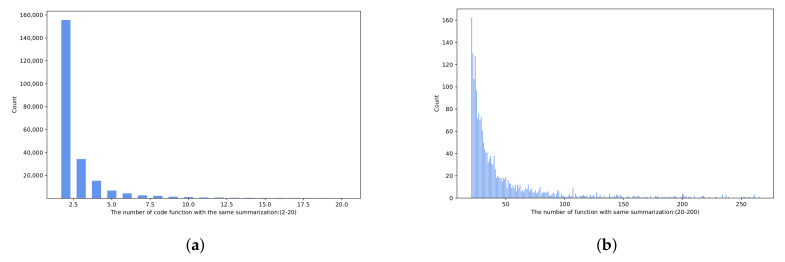
Java code scale distribution of the same code comment (color print).

**Figure 11 entropy-24-01372-f011:**
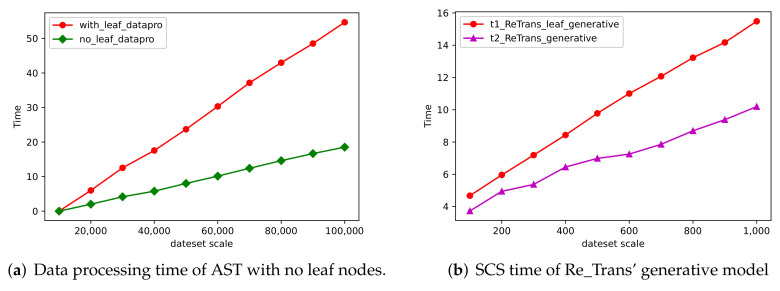
The efficiency of source code semantic extraction (color print).

**Figure 12 entropy-24-01372-f012:**
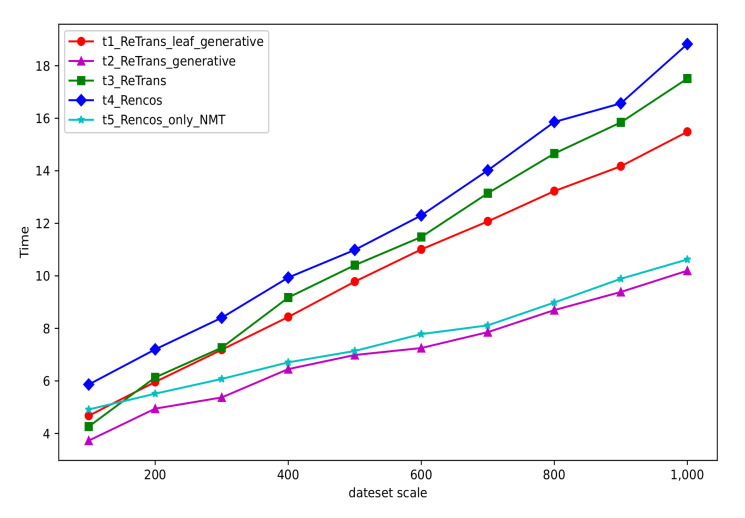
The time statistic of different models (color print).

**Figure 13 entropy-24-01372-f013:**
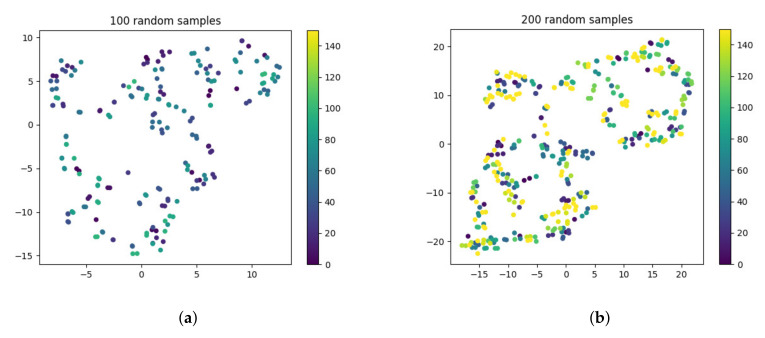
Distribution plot of 100/200 random samples (color print).

**Figure 14 entropy-24-01372-f014:**
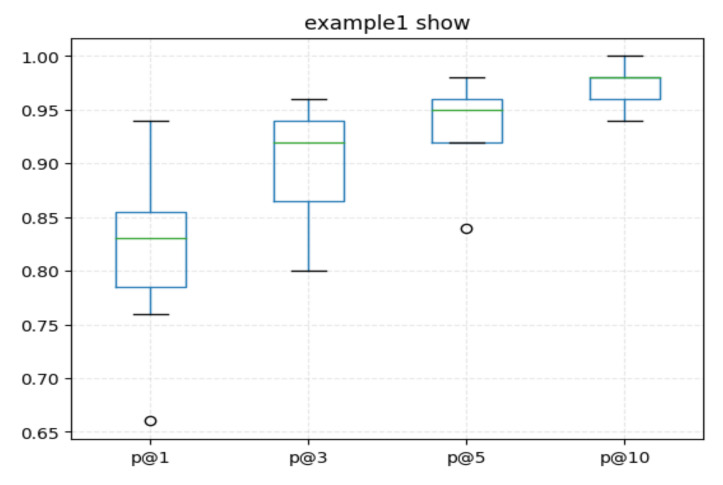
The time statistic of different models (color print).

**Table 1 entropy-24-01372-t001:** The length statistics of the dataset (numbers in the table represent the number of code’ words).

Function lengths	shortest	longest	average	<60	<70	<80
	1	100	29.7	87%	91%	95%
Comment lengths	shortest	longest	average	<9	<10	<11
	3	13	7.6	72%	81%	90%

**Table 2 entropy-24-01372-t002:** The comparison results of Re_Trans and Baselines (B-n represents BLEU-n).

Methods	B-1	B-2	B-3	B-4	ROUGE-L	METEOR
ast-attendgru [10]	37.24	22.14	14.32	11.06	39.68	19.31
graph2seq [31]	37.66	22.32	14.28	10.96	39.71	19.40
code+gnn+BiLSTM [24]	39.21	22.50	15.73	11.97	40.25	20.12
transformer-based [13]	39.67	24.96	16.21	13.77	40.87	21.05
Rencos [11]	36.63	21.61	15.11	12.30	39.70	19.17
Re2Com [32]	38.96	23.08	17.49	15.67	40.01	20.04
**Re_Trans(ours)**	**42.97**	**25.85**	**18.58**	**16.83**	**42.64**	**22.15**

**Table 3 entropy-24-01372-t003:** The ablation experiment results (B-n represents BLEU-n).

Methods	B-1	B-2	B-3	B-4	ROUGE-L	METEOR
Seq	35.58	22.36	15.22	10.86	37.63	18.97
ast_aug+Seq	38.99	23.74	16.76	15.32	40.01	20.26
**ast_aug+Seq_aug**	**42.97**	**25.85**	**18.58**	**16.83**	**42.64**	**22.15**
ast_leaf_aug+Seq_aug	43.01	26.30	19.22	16.30	42.68	22.01

## Data Availability

Not applicable.

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
