# Peer review of "Re_Trans: Combined Retrieval and Transformer Model for Source Code Summarization"

_entropy, 2022, doi:10.3390/e24101372_

Round 1
Reviewer 1 Report
The paper presents a potentially interesting combination of two techniques for source code summarization. Yet, it suffers from weaknesses as concerns the presentation style and lack of necessary explanations is several points, detailed below. Throughout the paper, readers are supposed to be quite familiar with the topic, which limits readability by general audience; the systematic (perhaps excessive) use of acronyms does not help, either. Some sentences in key points are vague, and leave readers in doubt about what authors actually mean - must be rewritten. Figures are often too small to be readable, and the systematic reference to colors in the explanation text makes it even harder to match explanation and figures.
The most critical point is perhaps the lack of a complete running example, which accompanies the reader from te beginning to the end of the paper: there are no examples at all of the type of output produced, both by ReTrans and by other systems. More examples could be provided in an Appendix or as supplementary material, but in the current manuscript the reader has no option to see even one single output of the system and therefore getting an idea of how it performs in comparison to other approaches. Numbers and statistics are helpful, of course, but some clear examples would greatly help and improve the paper.
My recommendation is that authors carefully revise the paper according to the suggestions below.
1) Introduction is more a related-work section than a real introduction: it would greatly help readability if related work could be moved to a separate section, and intro rewritten/extended so as to present the problem from a more general viewpoint, so as to make it clearly understandable also to readers who are not directly involved in the specific field, but could be interested in getting the main ideas and potential of this approach. Please consider reducing to a minimum the use of acronyms - there are really too many and they tend to be confusing.
- line 31: IR-based methods... (then next sentence starts with "it", does not match)
- line 40: "over reliance" - this is not clear, what do you mean? please clarify
- line 49 "adopted" - something missing in the sentence
- line 55 "The effect of.. is higher".. - that is? what do you mean? pls clarify
- line 63: "A RM"... but (it) requires..
- line 74: "more fully" - that is? what do you mean? pls clarify
- line 82: SOTA never introduced before, metrics should be shortly introduced (perhaps in a box) and provided with a reference
2) Section 2: initial paragraph is identical to lines 68-72, not necessary. Figures are too small (and there lacks a space between the "Figure" label and the number, not just here but in many points in the text, please fix); colors (lines 104-105) are not helpful when reading the paper printed in B/W, please adopts some different style (e.g. dotted lines, etc) to replace/in addition to colors to differentiate the type of lines. Moreover, figure 1 is divided in three sections named A,B,C but these identifiers are never mentioned in the text, which adopts a different naming style. Please be coherent and uniform.
- line 109: "simple text"- that is?
- line 111: "java toolkit" - that is? what is that, how is it made, ..? More gernally, the paragraph at lines 111-115 is too vague. You mention "non leaf nodes" but provide no explanation of what they represent and why (this is actually clarified in later sections, but the reader obviously cannot know: please either remove the sentence here, if not necessary, or add the necessary explanation, at least in terms of forward pointer to next sections)
-line 120: the ref to Figure 4 is wrong, should be Figure 2
- Figure 2/3 on page 4: the java example is really poor, an endless for cycle is really bad programming and should never be used as example, *unless* the expected output description is exactly to detect such kind of constructs: but since the output is not shown, the example per se is not helpful.
-line 128: the formula at the end of the line is unclear, but apart from that, there seems to be no reason to introduce it
- (avoid colors in description)
- lines 141-143 Emb1..Emb3, this part is totally unclear: no explanation of what these vectors are/represent, or what Emb is supposed to be. The next sentence (lines 144-146) is far too vague and unclear, too: please revise and extend it so as to make it clearly understandable by all readers.
- next page, lines 150-153: same paragraph repeated once again, not necessary. Please introduce and explain briefly what semantic similarity is and how it is measured (perhaps in a short side box). Also, next paragraph, please explain better what kind of Euclidean distance you use, if possible adding a short example, for the benefit of non-expert readers.
- section 2.4 is useless, merely repeats what has already been said before several times; instead, details about this could be done, and a runnig example, would be much more useful
- section 2.5: there is no clear explanation of how the discriminator is made and works, only its (obvious) function is actually defined. Authors should clearly present the discriminator component and explain its working approach, not just its expected output. The use of cosine distance is not introduced not motivated. At line 178, "sample < sample" is unclear (probably poor formatting of a <...> tuple ). At line 182, MLP acronym not explained - overll this section is "for experts only", the reader is not placed in condition to actually understand. Must be fixed.
3) In section 3, Figure 8, which is essential, is definitely too small. Moreover the caption of 8b says "Function" length distribution, while the text at lines 193-195 says "code length".. are these the same? Please be claer and coherent. Moreover, it is unclear why 70 should be the optimal input value: the motivation is too compact, please extend it and be clearer.
- next paragraph (lines 196-201) is not clear and should be rewritten/extended. Figures 9a/b/c are invisible: please enlarge it suitably. In comparison, Table 2 steals a lot of space, while providing just numeric details. Please consider re-tuning these aspects, or move Table 2 to an Appendix. Section 3.2 is only readable by expert readers.
- section 3.4, line 258: METROR should probably be METEOR. Apart from that, as mentioned above, I would suggest that a short introduction to such metrics is added for the benefit of non-expert readers (no formulas, just the idea behind them, and the key differences among them -- perhaps in a side box; actual definitions could be added in an Appendix, for the readersì convenience)
4) Line 269-270: RQ3 and RQ4 are not clear in English, please rephrase
- lines 277-279, sentence is unclear, authors refers to aspects (e.g. snippets) that they are the only ones to know, please make the sentence understandable by all readers.
- Same issue at lines 286-287: the sentence merely points to two references, please give the general point in explicit
- section 4.2: please introduce shortly what an ablation study is, so that the paper is self-contained. This section is very techical.
- section 4.3: in (1) and (2), I duggest to remove the starting sentence, which is merely a repetition of the title - just proceed to explanation; Figure 10 actually invisible, please enlarge suitably
Reviewer 2 Report
The authors propose a new source code summarization approach that combines retrieval-based and generative methods. Their idea is interesting and worthy of investigation, but the way the article was written needs to be improved.
First of all, you need to improve the used English language. Because of the language some paragraphs/phrases/sentences seem unfinished, and the reader has a hard time understanding your ideas. (An example is a paragraph on page 3 where you describe the steps of your approach, but there are also others, like: "However, IR-based methods over-reliance on identifier naming and the similar amount of source code in the 4dataset.". It seems to me that the phrase is missing something.)
Secondly, you must introduce all the acronyms used. For example, you never explain what AST stands for, nor GCN or MLP.
Thirdly, some of the images are too small and difficult to understand. Make them a little bit larger.
Fourthly, for the baseline presentation in Section 3.3, I would recommend including the names of the researchers proposing each method. Writing something like "(2019) ast-attendgru ..." is not appropriate. You should reformulate: Leclair et al. [10] proposed a method/approach called ast-attendgru that is ...".
Regarding the presentation of your approach, I would also recommend a few improvements:
1. You should include a more detailed example of a source code and the summarization obtained. You did include a small example of a method, but what is the end result for that method/piece of code? This way the reader will better understand your work and the purpose of your approach.
2. In Table 2, what do 60, 70, or 80 represent? Are they characters, lines of code, or something else? You do not specify in the text.
3. In Section 3.1, what do you mean by length? Whose length have you measured and in what (characters, line of code, etc)?
4. In Section 3.2, page 7, line 204, N, h, and dim should not be in parenthesis, as you explained their meaning outside of the parenthesis. Also, that sentence doesn't have a predicate. You need to reformulate.
Round 2
Reviewer 1 Report
Authors have addressed most of my previous remarks. Some minor details still need to be fixed, see list below.
Line 133: missing figure number
Line 132-133: to take care of the color issue, since the red lines is dotted, why not just mention this feature?
Current version: "For example, the green lines in the red box (aa) in Figure?? connect three sibling nodes of "Modifier”, “FormaParameter” and “ForStatement”. "
Proposed version: "For example, the green arrows in the red dotted box (aa) in Figure?? connect three sibling nodes of "Modifier”, “FormaParameter” and “ForStatement”. "
Line 152-3: Hence, each word in the source code sample has three feathers: token id, column position and row positiona
Line 162-3: and get the corresponding the embedding ..
Line 165-6: Let the word’s dimension is d --> Let the word dimension be d,...
Line 203: and introduce it through a running example that shown in Figure 6
Line 226: i ∈ [1, m] --> I see no "i" index actually
Line 259: The parameters of tansformer model
Line 306: that detailed --> detailed
Line 344: Ablation study is often to reduce --> is often used to reduce
